# Characterization of sound scattering layers in the Bay of Biscay using broadband acoustics, nets and video

Arthur Blanluet[1]*, Mathieu Doray[1], Laurent Berger[2], Jean-Baptiste Romagnan[1], Naig Le Bouffant[2], Sigrid Lehuta[1], Pierre Petitgas[1]

1 Unité Écologie et Modèles pour l'Halieutique, Ifremer, Nantes, France, 2 Service Acoustique Sous-marine et Traitement de l'Information, Ifremer, Brest, France

* arthur.blanluet@inra.fr

**Data Availability Statement:** Acoustical and biological data related to this work will be available on the IFREMER database SEANOE: https://doi.org/10.17882/62440.

## Abstract

Sound scattering layers (SSLs) are observed over a broad range of spatio-temporal scales and geographical areas. SSLs represent a large biomass, likely involved in the biological carbon pump and the structure of marine trophic webs. Yet, the taxonomic composition remains largely unknown for many SSLs. To investigate the challenges of SSL sampling, we performed a survey in a small study area in the Northern Bay of Biscay (France) by combining broadband and narrowband acoustics, net sampling, imagery and video recordings. In order to identify organisms contributing to the observed SSLs, we compared measured frequency spectra to forward predicted spectra derived from biological data. Furthermore, to assess the confidence in SSL characterization, we evaluated uncertainties in modeling, acoustical and biological samplings. Here, we demonstrate for the first time that SSL backscattering intensity in the Bay of Biscay can be dominated in springtime by resonant gas bearing organisms below 100 kHz, namely siphonophores and juvenile fishes and by pteropods at higher frequencies. Thus, we demonstrate the importance of broadband acoustics combined to nets, imagery and video to characterize resonant backscatterers and mixed mesozooplankton assemblages.

## Introduction

Sound Scattering Layers (SSLs) are routinely observed with active acoustic devices in a great variety of ecosystems and over wide depth ranges in the global ocean [1–4].

Deep Scattering Layers [5] inhabiting the mesopelagic zone worldwide, are *e.g.* known to perform daily the largest migrations on earth [6] and their fish component might dominate the world total fishes biomass [4, 7]. SSLs might hence play an important role in the biological carbon pump [8] and in the structure of marine trophic webs [9, 10].

SSLs are generally composed of mixed species assemblages of great diversity in taxa, body size and acoustic backscattering. Mesozooplankton (0.2-20 mm, [11]), macrozooplankton (> 20 mm, mainly shrimp and gelatinous zooplankton, [11]) and micronekton (20-50 mm,

**Funding:** The Arthur Blanluet grant was co-funded by the Conseil Régional des Pays de la Loire and IFREMER.

**Competing interests:** The authors have declared that no competing interests exist.

mainly small mesopelagic fish and other fish juveniles, [12]) have hence been reported to significantly contribute to SSLs backscatter. If SSLs are easy to observe with active acoustic devices, their taxonomic and size diversity makes them difficult to sample and identify [1]. Uncertainty of the species composition in the SSL's generally hinders the comprehension of their ecological functions. To overcome SSLs composition identification issues, we must to understand: i) the limits of each sampling method, and ii) how these methods can complement one another.

Net sampling is a traditional way to identify organisms composing SSLs, but it is discrete in time and space and highly dependent on target catchability, as some organisms actively avoiding nets [13]. Furthermore, nets generally destroy fragile organisms such as jellyfish. Optical sampling is a good way to evaluate the presence of fragile organisms, albeit with low range and small sampling volumes. Still, like nets, optical sampling is affected by avoidance and behavior of organisms [14]. Acoustic sampling is continuous over various spatial scales with good resolution. It is however an indirect sampling method with low taxonomic resolution [15] and sensitivity to shadowing issues [16]. Additionally, the backscattering intensity of different types of scatterers is extremely variable [16] and the strongest backscatterers are not necessarily the most abundant. Until recently, most SSL acoustic studies were conducted using narrowband, multi-frequency acoustics [1, 14, 17]. However, the emergence of broadband acoustics [18] as standard monitoring tools has the potential to improve species identification based on their broadband frequency spectra [19–21].

Ubiquitous dense SSLs have been routinely observed during the PELGAS (PELagic GAScogne) multidisciplinary integrated survey which has taken place in springtime since 2000 [22]. Still to date, the composition of these layers remained largely unknown. Interestingly though, the presence of small gas-bearing targets in the Bay of Biscay water column was demonstrated by the detection of distinct acoustic resonance peaks [23, 24].

Resonance occurs when some specific types of targets (as gas bubbles) increasingly vibrate upon insonification at a given frequency [25, 26]. This natural frequency of the target depends on the target size and its physical properties as well as the one of the surrounding medium. This resonance can induce a dramatic increase in the backscattering intensity of small gas-bearing organisms (e.g. physonect siphonophores or small swimbladdered fish), relatively to their size [4, 14]. Previous studies suggested that Biscay dense SSLs were mainly composed of gaseous resonant target, as fish larvae, gelatinous gas-bearing siphonophores or phytoplankton [23, 27]. But no biological sampling had supported these conclusions.

In this study, we investigated the composition of SSLs observed on the continental shelf of the northwestern Bay of Biscay. We combined broadband acoustic data with alternative sampling methods, including optical and net recordings. Our goals were to (i) evaluate the input of broadband acoustics in SSL studies, (ii) improve the identification of organisms dominating the SSL backscattering by using a forward approach [28], and (iii) investigate the various sources of uncertainty associated to each SSL characterization method.

## Materials and methods

### Sampling

The PELGAS survey take place every year with the French Navy (Marine Nationale) permission. The 2016 permit number is: IF 14 CECLANT/OPS/OPSCOT. Sampling took place on the R/V Thalassa at the end of the PELGAS2016 survey [29] in the Northern Bay of Biscay, between the 27th and the 28th of May. SSLs were sampled over a diel cycle in one area off south Brittany (Fig 1). Here, we present the daytime biological sampling and the corresponding acoustic data that were collected at speeds between 2 and 3 knots.

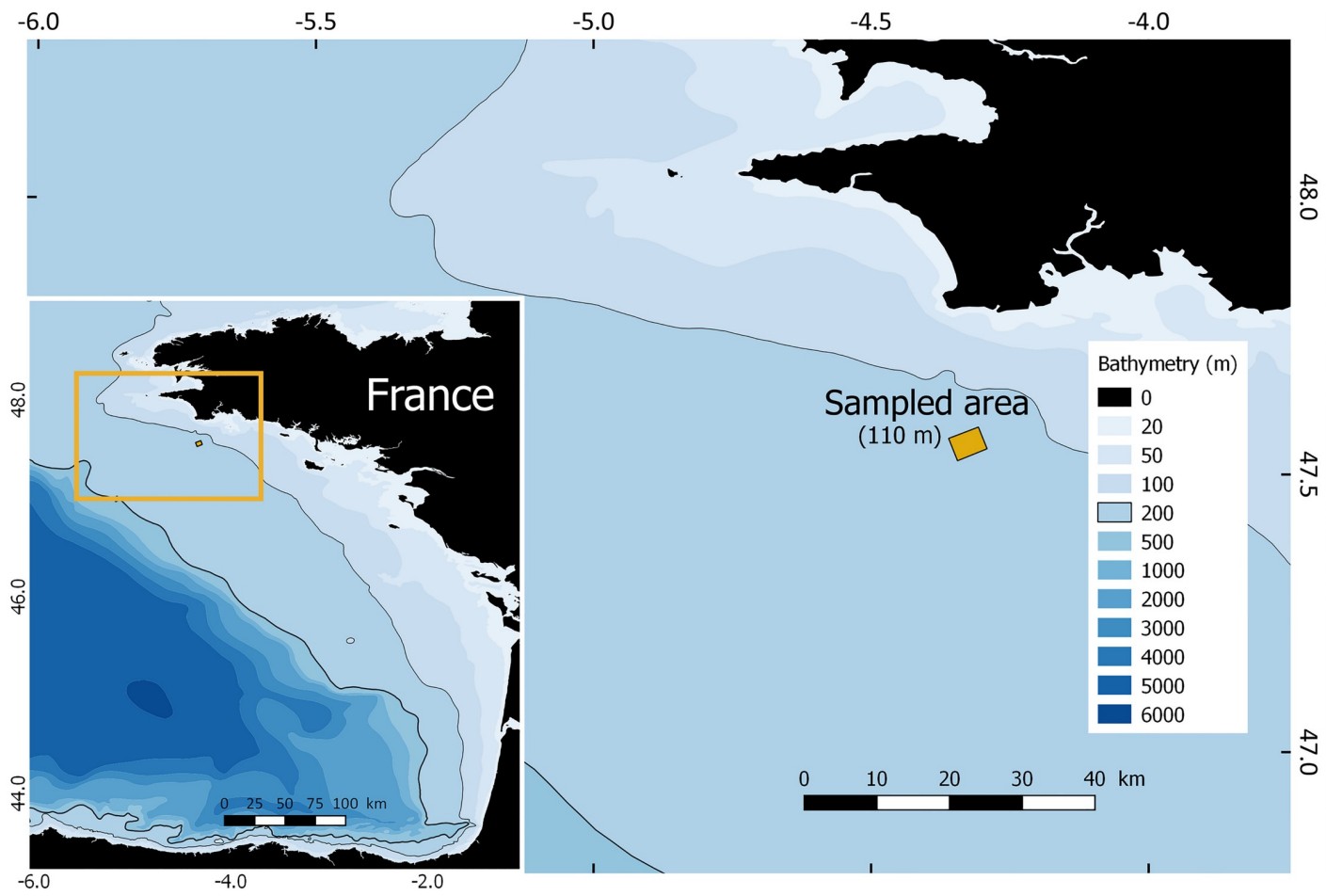

**Fig 1. Study area in the north-western Bay of Biscay (average depth: 110 m).**

**Acoustic sampling.** R/V Thalassa was equipped with two hull-mounted Simrad EK60 split-beam narrowband (*Continuous Wave* pulse, CW) echosounders transmitting at 18 kHz and 38 kHz and four hull-mounted Simrad EK80 split-beam echo-sounders operated in broadband mode (*Frequency Modulated* pulse, FM) transmitting from 47 kHz to 420 kHz in plural upward sweeps (Table 1). The 6 echosounders were set to ping sequentially at 2 pings per second (PPS) (i.e. one ping every 3 seconds in each band) to avoid cross-channel interferences. The transducers were located at 6m below sea surface. Acoustic data collected above 10 m deep were discarded to exclude the 18 kHz echosounder post transmit ringing and to ensure that acoustical data were collected within the far field of each echosounders.

Calibration was performed using the SIMRAD EK80 software (version 1.11.1) with default calibration settings [30] (see details in S1 Appendix). Frequency regions with noise spikes or low backscattering sensibility were removed from any further analysis (e.g. 240-260 kHz, part of 200 kHz and 333 kHz bandwidth when noise peak at depth interfered with the signal, S1 Fig in the Supporting Informations). Due to its limited range, the 333 kHz band was not used in the following cluster analysis.

EK80 broadband raw data were pre-processed using customized MATLAB codes developed at Ifremer to obtain measurements of frequency-dependent volume backscattering strength

**Table 1. Echosounders settings.**

| Transducer | ES18-11 | ES38B | ES70-7C | ES120-7C | ES200-7C | ES333-7C |
|---|---|---|---|---|---|---|
| Signal | CW | CW | FM | FM | FM | FM |
| Ramping[a] | NA | NA | Fast | Fast | Fast | Fast |
| Bandwidth[b] (kHz) | 18 | 38 | 47-90 | 95-160 | 180-240 | 280-420 |
| Pulse duration[b](ms) | 1.024 | 1.024 | 2.048 | 2.048 | 2.048 | 2.048 |
| Transmit power[b] (W) | 2000 | 2000 | 600 | 200 | 90 | 40 |
| Nominal opening[c] | 11˚ | 7˚ | 7˚ | 7˚ | 7˚ | 7˚ |
| Calibration sphere size (mm) | 38.1 | 38.1 | 38.1 | 38.1 | 22.0 | 22.0 |
| Calibration gains | PELGAS 2016 | PELGAS 2016 | PELGAS 2017 | PELGAS 2017 | IFREMER Brest Tank | IFREMER Brest Tank |

[a] [18]

[b] Value recommended by the manufacturer

[c] In FM mode, beam angles vary with frequency. We assumed that variable beam angle had no impact on the SSLs volume backscattering strength at frequency f assuming a homogeneous target distribution.

($S_v$, $dB \, re \, 1m^{-1}$, [31]), following the 2016 USA–Norway EK80 Workshop recommendations [18]. These code routine are implemented in MOVIES3D freeware [32]. An important step in broadband data post-processing is the choice of the Fast Fourier Transform (FFT) window width (*i.e.* the amount of data points used to compute frequency spectra) [21]. Reducing the FFT window length increases the spatial resolution, but decreases the frequency resolution. We used a frequency resolution of 2 kHz for the spectrum analysis, corresponding to an effective temporal resolution of 0.34 m. A 0.6-Tukey shading window was applied to the data before applying the FFT to reduce the leakage of spectral analysis. Individual $S_v$ spectra were obtained by an echo-integration of 1.5 m vertically and 5 pings horizontally, which covered a horizontal distance of approximately 15-23 m. This integration volume contained approximately 20 points, and was a trade-off between a stable frequency response and spatial resolution (ideally around 100 points, [33]).

**Biological sampling.** Biological sampling was performed with a Methot Isaac Kidd (MIK) net for specimens over 2 mm and with a Multinet (Hydro-Bios, Germany) for individuals under 2 mm. Net characteristics are presented in Table 2, using [34] tow type nomenclature. Net tow depths and durations are presented in S1 Table in the Supporting Informations. All nets were equipped with a depth sensor. The multinet was equipped with 5 nets and an opening/closing system triggered by a pressure sensor. The MIK net was equipped with a GOPRO Hero 4 video camera encased in a pressure container fitted with two 2000 lumens white LED spotlights. The video camera system was used to detect the presence of fragile centimetric organisms that would have been destroyed by the net. The camera was oriented towards the

**Table 2. Characteristics of the nets used for biological sampling.**

| | Mesh ($\mu m$) | Mouth shape | Mouth surface ($m^2$) | Net length (m) | Sampled volume estimation | Type of deployment[a] |
|---|---|---|---|---|---|---|
| Multinet 5x[b] | 335 | square | 0.25 | 2.50 | flow-meter | Multiple (5) closing nets, oblique tow |
| MIK net | 1600/500 | cylindrical | 3.14 | 13 | surface × distance | oblique-horizontal tow |

[a] Based on [34] tow type nomenclature

[b] Multiple closing gear with 5 nets equipped with opening/closing system triggered by a pressure sensor

MIK codend to ensure that all organisms entering the net would pass in the camera field of view.

Additionally, we performed Conductivity-Temperature-Density (CTD, Seabird, USA) vertical profiles. The CTD probe was equipped with a fluorometer (Seabird, USA), a turbidimeter (Campbell Scientist, USA) and an oxygen sensor (Seabird, USA). The CTD profiles are presented in S3 Fig in the Supporting Informations.

We sampled two different depth layers, corresponding to the depth ranges of the densest and most structured SSLs observed in real time on pulse compressed echograms (Fig 2): i) a surface layer (from 10 to 24m depth, located at the thermocline/halocline level) and ii) a deep layer (from 70 to 96m depth). Each layer was sampled by two MIK tows and two multinet net tows.

Micronekton and gelatinous zooplankton caught by the MIK nets were identified and measured immediately after collection, or later in the laboratory after formalin fixation (4%). Multinet mesozooplankton samples were also imaged immediately after collection using the ZooCAM in-flow imaging system [35]. Mesozooplankton between 1 and 3 mm from the MIK samples were imaged using a ZooScan [36] after formalin fixation (4%) in the laboratory. The Ecotaxa web application [37] was used to identify, count and archive images originating from

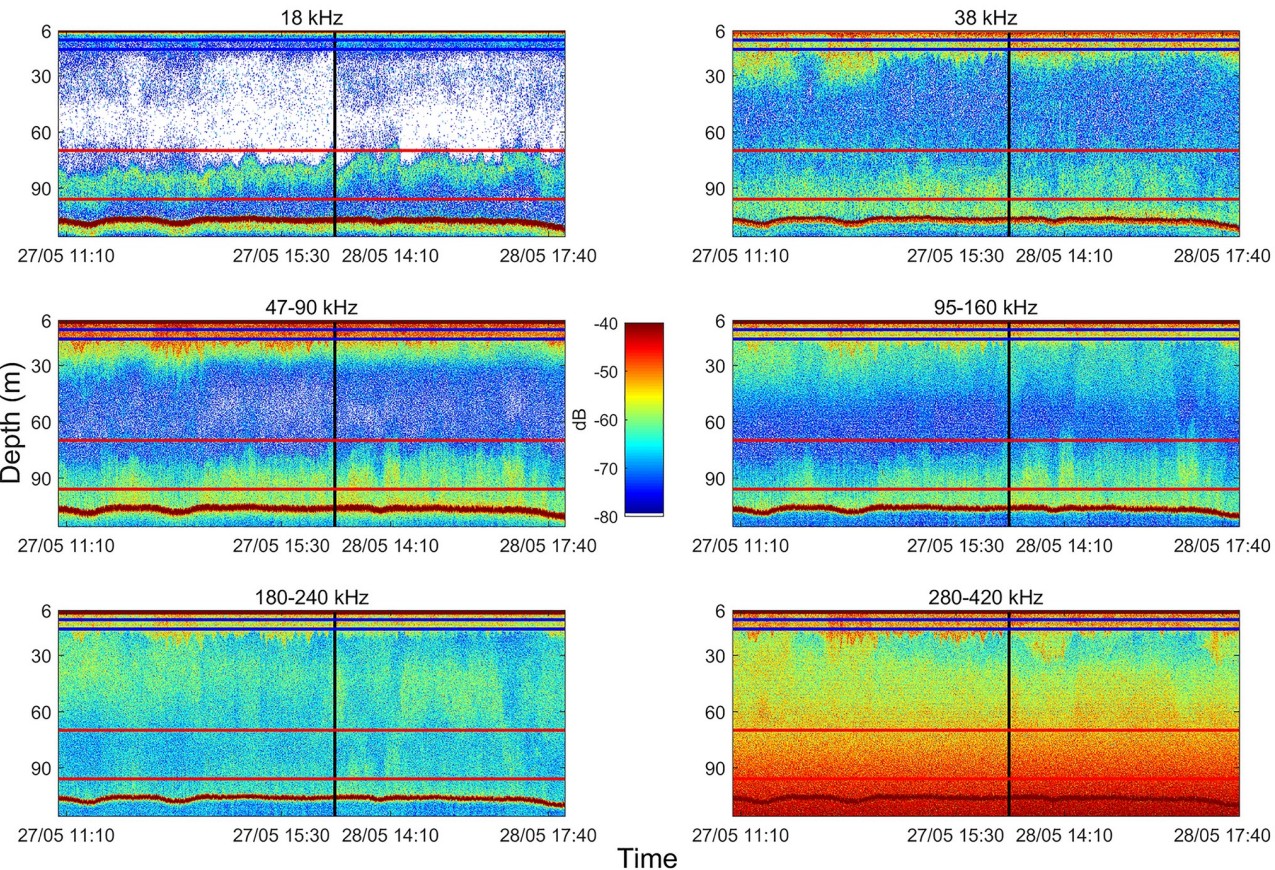

**Fig 2. Pulse compressed $S_v$ echograms recorded during net tows.** The vertical blackline separate data collected on May, 27th and 28th. Blue and red horizontal lines represent the boundaries of surface and deep sampled layers, respectively.

both instruments. The video footage recorded during MIK tows were visually analysed to detect the presence of large gelatinous organisms, such as jellyfish or siphonophores.

Density and size of fish and euphausiids larger than 2 mm were estimated from MIK samples. Density and size of mesozooplankton (copepods, pteropods) and euphausiids smaller than 2 mm were estimated from Multinet samples.

Large physonect siphonophore bodies were visually counted on GOPRO videos. As they are always composed of colonial bodies with one single gaseous inclusion (the pneumatophore), densities of pneumatophores were estimated by dividing organism counts by the net sampling volume. We assumed that pneumatophores larger than 0.3 mm found in the catches came from large siphonophore bodies observed on the videos recorded during the same tow. Siphonophores bearing pneumatophores smaller than 0.3 mm were too small to be observed on videos. Their densities were estimated based on the number of pneumatophores found in the Multinet samples which we processed using imagery. A small number of other weak acoustic scatterers (small jellyfish, comb jelly and swimming crabs) were also caught but were not included in the forward approach.

## Forward approach

Using the forward approach [28, 38], a predicted Volume backscattering Strength ($S_v$, $dB$ $re$ $1m^{-1}$, [31]) at a given frequency (f) is estimated with individual Target Strength (TS, $dB$, [31]) models, parameterized by the biological samples (size and shape). Predicted $S_v(f)$ were then compared to our *in situ* $S_v(f)$ averaged in the area where biological samplings were performed.

The predicted $S_v(f)$ ($S_{v\ predicted}(f)$) were calculated for each depth layer as the sum in natural scale of the frequency spectrum $S_{vj}(f)$ of each sampled taxa $j$ (Eq 1):

$$S_{v\ predicted}(f) = 10 * log_{10} \sum_{j=1}^{N\ taxa} 10^{S_{vj}(f)/10} \tag{1}$$

$S_{vj}(f)$ was calculated as:

$$S_{vj}(f) = 10 * log_{10} \sum_{i=1}^{n\ length\ class} 10^{(TS_i + 10 * log_{10}(D_i))/10} \tag{2}$$

Where $TS_i$ represents the modeled Target Strength of length class $i$ in $dB$ $re$ $1m^2$ and $D_i$ the sampled density of the organisms (in individual by $m^3$) belonging to the $i^{th}$ length class (Eq 2). Models and parameters used for each organism are presented in the next section.

The measured $S_v(f)$ ($S_{v\ measured}(f)$) of each layer were calculated as the median of the $S_v(f)$ in echo-integration cells along the sampling net trajectory. These cells were defined as 2 echo-integration cells located below the top of the net. Uncertainty in the predicted and measured $S_v(f)$ curves are considered below.

## Scattering models

**Fluid-like organisms.** A scattering model based on the distorted-wave Born approximation (DWBA) [39] was used to model the backscattering of Fluid-like (FL) organisms [16], for all frequencies and averaged over a normal distribution of orientation angles. A prolate spheroid model was used for copepods and an uniformly-bent and tapered cylinder model for shrimp-like organisms and fish lacking swimbladders. Model parameters are summarized in Table 3. Euphausiids model parameters come from [2], copepods model parameters from [40] and fish without swimbladders model parameters from [41].

**Table 3. Scattering model parameters used for Fluid-like organisms.** SD: Standard Deviation.

| Organism (*Scattering Model*) | Length(L)-to-width(2a) ratio (L/2a)[a] | Orientation (Mean, SD) | Density contrast (g) | Sound speed contrast (h) |
|---|---|---|---|---|
| Euphausiids and Decapod Shrimp (*DWBA uniformly-bent cylinder*) | 5 | N(20,20)[b] | 1.016[b] | 1.019[b] |
| Copepods (*DWBA prolate spheroid*) | 2.55 | N(90,30)[b] | 0.949[c] | 0.995[c] |
| Fish without Swimbladders (*DWBA uniformly-bent cylinder*) | 4 | N(0,30) | 1.01[d] | 1.025[d] |

[a] is the radius of the spherical section of the cylinder or of the prolate spheroid

[b] [2]

[c] [40]

[d] Value reported by [41] for *Scomber scombrus*

In the case of net samples processed with imagery, the length of FL organisms was estimated as the major axis of their silhouette area observed on imagery pictures. The radius *a* of an organism's cross section was estimated as the semi-minor axis of the planar projection of the model shape (rectangle for a cylinder, ellipse for a spheroid) whose area was equal to the silhouette area. The mean ratio L/a of body Length (L) over cross section radius (a) was calculated on a subset (*n* > 30 by scatterer types) of imagery pictures without flexion or cut body part. In the case of organisms processed manually, the mean ratio L/a was estimated based on a subset of a dozen individuals.

**Elastic shell organisms.** Thecosome gastropods (common name pteropods) produce relatively high backscattering in geometric scattering compared to FL organisms of similar size [16]. Pteropods are caracterized by a hard aragonite shells, with a large discontinuity: the opercular opening [2]. Based on the review of Elastic Shell (ES) scattering models by [2], we selected a high-pass dense-fluid sphere model with an empirically derived reflection coefficient (*R* = 0.5) to predict the backscattering intensity of ES organisms [16].

Pteropods sampled in the nets were measured by plankton imagery. Their cross section radius *a* was estimated as the radius of a disc of the same area as the organism's silhouette image.

**Gas-Bearing organisms.** We used a gaseous prolate spheroid model [42], supplemented with [26]'s damping equation (see details of the model in S2 Appendix) to predict the backscattering of Gas Bearing (GB) organisms [16]. This model is valid for Rayleigh scattering, i.e. for $ka < kb < 0.1$; where $a$ and $b$ are respectively the semi-minor and semi-major axes of the spheroid in meter and $k$ is the wave number ($m^{-1}$).

Swimbladdered fish viscosity and surface tension parameters come from [43] and [44]. Swimbladder semi-minor axis *a* and length-to-width ratios (L/2a) of transparent fish (juveniles or larvae of *Crystallogobius linearis* and *Carapus acus*) were visually measured in a sample of thirty specimens, all of which were in good condition. A mean ratio relating fish body length to swimbladder length was derived based on these measurements. This ratio was used to predict the swimbladder cross-section of damaged individuals.

When direct measurements were impossible, we approximated the fish body volume as a prolate spheroid, with the fish length as the major axis and 1/6 of the fish length as the minor axis (based on the mean of all visual measurements). Following the estimations and measurements of [45], [46] and [47], swimbladder volume was then estimated as 2.5% of the body volume, using an empirical swimbladder length-to-width ratio of 1.5. These values were in line with the swimbladder to body volume estimations made on transparent fishes.

Pneumatophores of physonect siphonophores displayed a pneumatophore mean length-to-width ratio (L/2a) of 2.35, regardless of their size. As no specific value of viscosity and surface

**Table 4. Scattering model parameters used for Gas Bearing organisms.**

| Taxon | Length-to-width ratio ($L/2a$) | Viscosity ($\eta$, in $Pa.s^{-1}$) | Surface tension ($\tau$, in $N.m^{-1}$) |
|---|---|---|---|
| *Crystallogobius linearis* | 1.63 | 1[a] | 200[a] |
| *Carapus acus* | 2.75 | 1[a] | 200[a] |
| Other fish | 1.5 | 1[a] | 200[a] |
| Siphonophores pneumatophore | 2.35 | 0.1[b] | 15[b] |

[a] [43] and [44]

[b] No specific value for viscosity/surface tension of the pneumatophores, they were fixed arbitrarily

tension could be found in the literature for siphonophores pneumatophores, those parameters were arbitrarily set to 0.1 $Pa.s^{-1}$ and 15 $N.m^{-1}$, respectively. Those values were chosen to lay between values reported for fish flesh (1 $Pa.s^{-1}$ and 200 $N.m^{-1}$, from [44]) and water (0.0013 $Pa.s^{-1}$ and 0.075 $N.m^{-1}$) and to provide the best fit between the measured and predicted $S_v(f)$. Parameters used in GB organisms scattering models are presented in Table 4.

## Uncertainty analysis

**Uncertainty analysis of scattering models.** Model parameters used in this study to estimate organisms backscatter were not specific to the Bay of Biscay. We performed an uncertainty analysis for each scattering model to assess the reliability of model predictions, when accounting for uncertainty in model parameters derived from the literature. Reference values for parameters other than length (that was measured on all organisms) are presented in Tables 3 and 4. The standard deviations of the parameter distributions were based on the range of the literature. Alternative values for the parameters were drawn from statistical distributions centered on reference values. The mean, standard deviation and quantiles of these model parameter distributions are presented in Table 5. The uncertainty on the FL organism's orientation was represented by using a mean angle distribution in the DWBA model (Table 3) and was therefore not included in the uncertainty analysis.

The uncertainty analysis consisted of 1000 model simulations run in each 0.1 $\mu m$ scatterer size class. The parameter values were drawn from the distributions presented in Table 5, using Latin Hyper Cube sampling [48]. This near-random sampling method ensured that each parameter was sampled according to its distribution, and that the parameter space was homogeneously explored. Confidence intervals around scattering model predictions were defined as the 5% and 95% quantiles of the simulation results.

**Variability of *in situ* frequency responses.** The SSLs spatial and frequential heterogeneity along net trajectories was estimated to assess the representativity of biological sampling. Clusters of echo-integration cells were defined on echograms based on $S_v(f)$ and net tracks were superimposed on the segmented echograms. To assess frequency spectra heterogeneity along each net track, we compared median frequency spectra computed along net tracks to the median $S_v(f)$ of each cluster. Echogram segmentation was performed using Expectation Maximization (EM) clustering [49], following [50] recommendations. Data used in EM clustering were standardized by the mean of the $S_v(f)$ spectrum weighted by each transducer bandwidth, as follows:

$$S_{v\ standardised} = S_v(f) - 10 * log_{10}\left(\frac{\sum_{n=1}^{N\ transducers} \overline{s_{vn}}}{N\ transducers}\right) \quad (3)$$

**Table 5. Statistical distributions for the model parameters used in the uncertainty analysis.** $\mu$: mean, SD: standard deviation, 5% and 95% quantiles of the distribution: Confidence Interval (0.05-0.95) the, $L/2a$: length/width ratio, $L/a$ length/half-width ratio.

| Parameters | Distribution | $\mu$ | SD | Median | CI(0.05-0.95) |
|---|---|---|---|---|---|
| Siphonophores, Ye modified model | | | | | |
| $L/2a$ | Normal | 2.35 | 0.3 | 2.35 | 1.86-2.84 |
| Viscosity ($\tau$) | Log-normal | -2.3 | 1 | 0.1 | 0.019-0.52 |
| Surface tension ($\eta$) | Log-normal | 2.7 | 1 | 14.88 | 2.87-77.1 |
| Swimbladdered fish, Ye modified model | | | | | |
| $L/2a$ | Normal | Species dependent[a] | 0.2 | | Species dependent[a] |
| Viscosity ($\tau$) | Log-normal | 0 | 1 | 1 | 0.19-5.18 |
| Surface tension ($\eta$) | Log-normal | 5.3 | 1 | 200.3 | 38.7-1037.8 |
| Copepods, DWBA model | | | | | |
| $L/a$ | Normal | 5.5 | 1 | 5.5 | 3.86-7.15 |
| Density contrast (g) | Normal | 0.96 | 0.0075 | 0.96 | 0.948-0.978 |
| Sound speed contrast (h) | Normal | 0.99 | 0.0075 | 0.99 | 0.978-1.002 |
| Euphausiids, DWBA model | | | | | |
| $L/a$ | Normal | 10 | 1.5 | 10 | 7.53-12.47 |
| Density contrast (g) | Normal | 1.016 | 0.0075 | 1.016 | 1.004-1.028 |
| Sound speed contrast (h) | Normal | 1.019 | 0.0075 | 1.019 | 1.007-1.031 |
| Fish without swimbladder, DWBA model | | | | | |
| $L/a$ | Normal | 8 | 1.5 | 8 | 5.53-10.47 |
| Density contrast (g) | Normal | 1.01 | 0.0075 | 1.01 | 1.000-1.022 |
| Sound speed contrast (h) | Normal | 1.025 | 0.0075 | 1.025 | 1.0127-1.037 |
| Pteropods, High pass dense-fluid sphere model | | | | | |
| Reflection coefficient (R) | Log-normal | -0.69 | 0.4 | 0.50 | 0.26-0.97 |

[a]Fish swimbladder length-to-width ratios ($L/2a$) per species are presented in Table 4

with $\overline{s_{vn}}$ the $s_v$ mean value (in natural scale) over the whole bandwidth of the transducer $n$:

$$\overline{s_{vn}} = \frac{\sum_{b=fmin}^{fmax} 10^{S_{vb}/10}}{f_{bandwidth}} \qquad (4)$$

with $Sv_b$ the backscattering at the $b$ frequency of the transducer $n$ bandwidth, $f_{bandwidth}$ the number of frequency bins within the bandwidth, $fmin$ and $fmax$ the bandwidth boundaries of transducer $n$, presented in Table 1.

We hypothesized that layers were stratified horizontally (coherent with the echograms on Fig 2), thus each "k" cluster was initialized by the mean (in natural scale) $S_v(f)$ of "k" equal horizontal bands stratifying the layer. The number of clusters used in the final segmentation was set to maximize the diversity of $S_v(f)$ spectra shapes while minimizing the number of clusters.

## Results

### Biological sampling

The density, mean length and Standard Deviation (SD) of sampled organisms in each zone and layer are presented in Table 6. Illustrations of the sampled main scatterers can be find in the S2 Fig in the Supporting Informations. CTD profiles are presented in the S3 Fig in the Supporting Informations.

**Table 6. Composition and length of each layer main scatterers.** SD: Standard Deviation.

| | Surface layer | | Deep layer | |
|---|---|---|---|---|
| **Dominant scatterers** | **Mean length (SD) (mm)** | **Density (*ind/m³*)** | **Mean length (SD) (mm)** | **Density (*ind/m³*)** |
| Euphausiids | 4.26 (2.22) | 14.2 | 13.7 (9.15) | 1.53 |
| *Limacina sp.* | 0.75 (0.12) | 5358 | 0.71 (0.15) | 126 |
| Copepods | 1.28 (0.37) | 1403 | 1.41 (0.52) | 126 |
| Swimbladdered fish[a] | 1.53 (0)[b] | NA[b] | 0.52 (0.15) | 0.0065 |
| Fish without swimbladder *Argentina sphyranea* | | | 21.5 (3.89) | 0.012 |
| Physonect siphonophore pneumatophore[a] | 0.27 (0.11) | 8.32 | 1.23 (0.31) | 0.48 |
| Jellyfish (Pandeiidae)[c] | 30.1 (10) | 0.033 | 24.47 (13.38) | 0.006 |
| Swimming crabs *Polybius henslowii* [c] | | | 31.75 (3.54) | 0.0015 |
| Comb jelly *Pleurobrachia pileus* [c] | | | 13.6 (7) | 0.0022 |

[a] Fish swimbladder length-to-width ratios (*L/2a*) per species are presented in Table 4

[b] Only one individual sampled, not modeled

[c] Not modeled due to low densities and weak backscattering

We found Pteropods (*Limacina sp.*) and Copepods in high density near the surface (5300 $ind.m^{-3}$ and 1403 $ind.m^{-3}$ respectively). High densities of small euphasiids (4.26 mm of mean length, 14 $ind.m^{-3}$) were also found in the surface layer, while large euphausiids were sampled in the deep layer (14 mm of mean length, 1.5 $ind.m^{-3}$).

We sampled two sizes of physonect siphonophore pneumatophores: small siphonophores (mean pneumatophore length 0.3 mm) present in the surface layer (8.3 $ind.m^{-3}$) and larger siphonophores (mean pneumatophore length 2 mm) caught at a lower density in the deep layer (0.5 $ind.m^{-3}$). The body length of large siphonophore was estimated at about 1 meter, based on video recordings, using the known MIK net opening as scale. The body size of small siphonophores was unknown as they were not observed by the video camera.

We only caught one fish in the surface layer, a *Ciliata mustela* larvae. In the deep layer we mostly caught non-swimbladdered juveniles of *Argentina sphyranea* [51] which were more abundant in the catch than swimbladdered fish species (0.012 $ind.m^{-3}$ and 0.0065 $ind.m^{-3}$ respectively). The swimbladdered fish were mostly composed of *Crystallogobius linearis* juveniles (50% in number) and *Carapus acus* larvae (20%). The remaining 30% was composed of juvenile gobiidae, blenniidae, triglidae and gadidae.

Lastly, we found a small amount of jellyfish (Pandeiidae spp.), *Polybius henslowii* swimming crabs and *Pleurobrachia pileus* comb jellies in the samples (Table 6).

## Sound scattering layers internal variability

The results of the EM clustering of echo-integration cells within the layers sampled by the nets are presented in Fig 3.

The surface layer was split into two vertically stratified clusters (Fig 3(a)). The $S_v(f)$ spectra of both cluster displayed a similar shape above 100 kHz with a decrease in the overall backscattering intensity with depth. In contrast, both spectra differed between 38 and 95 kHz, with a single broad peak around 80 kHz for cluster 2, and two peaks at 38 and 80 kHz for cluster 1. Multinet U0354 and MIK net tows each sampled exclusively one cluster, cluster 1 and 2, respectively. However, the multinet U0341 sampled clusters 1 and 2 equally.

The deep layer was separated into three vertically stratified clusters, whose mean backscattering intensity increased with depth (Fig 3(b)). The median $S_v(f)$ spectrum of each cluster

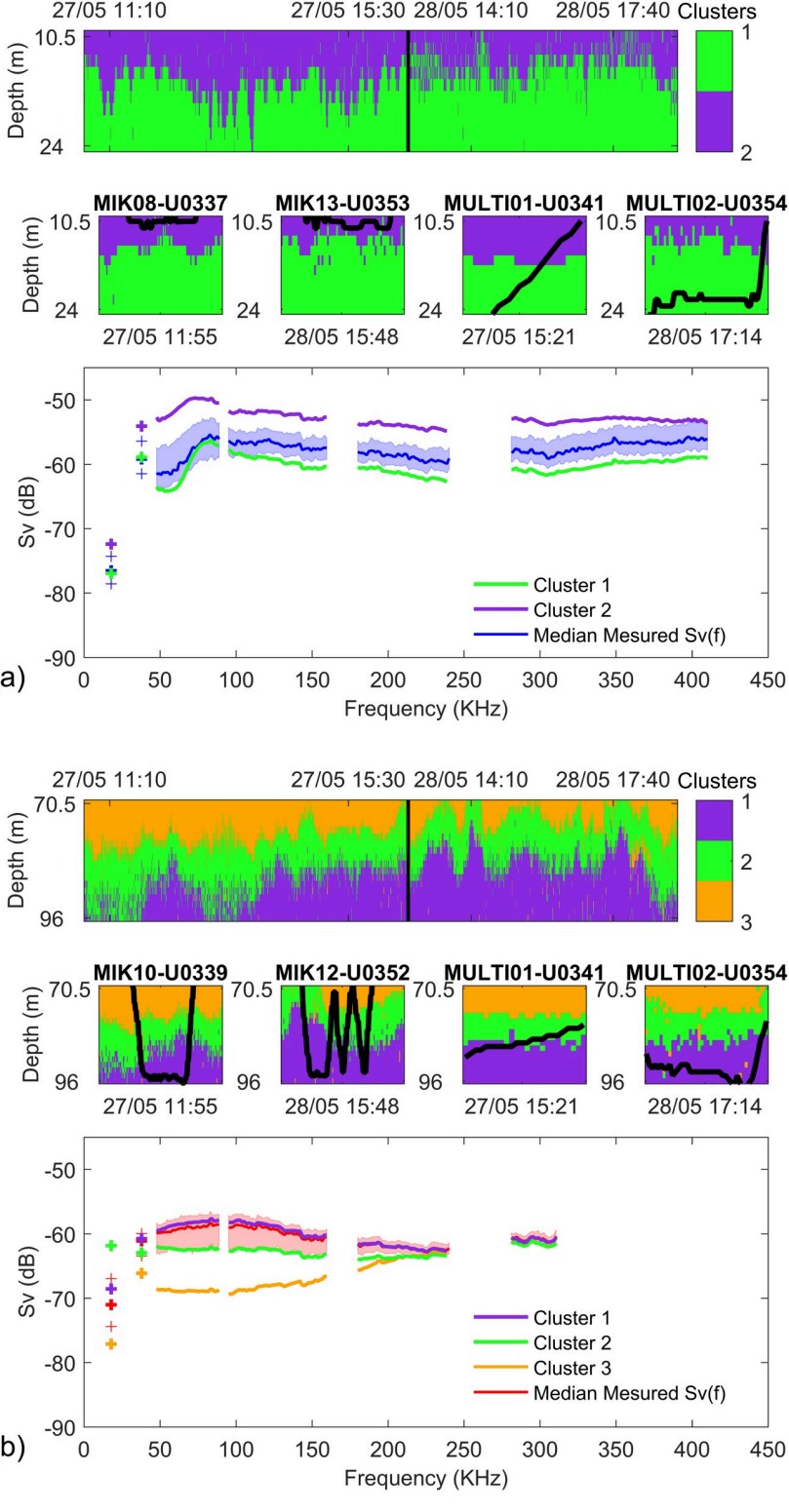

**Fig 3. EM clustering results for (a) the surface layer and (b) the deep layer.** Upper panels: echo-integration cells belongings to cluster; left of vertical black line: data collected on May, 27[th], right of black line: data collected on May, 28[th]. Middle panel: net trajectories (black lines). Lower panel: median volume backscattering strength spectra at frequency f ($S_v(f)$) in the whole depth layer (red and blue curve with 25% and 75% quantiles confidence intervals) and in each cluster.

displayed specific shapes: a peak at 38 kHz then a rise for cluster 3; a flat spectrum for cluster 2 and a bump between 18 and 150 kHz for the cluster 1, followed by a flat curve beyond 150 kHz. Most net tows sampled the deepest cluster (1), with the exception of the W-shaped MIK U0352 tow that sampled the whole layer and the multinet U0341 that sampled the interface between cluster 1 and cluster 2.

### Forward approach

Predicted and measured $S_v(f)$ curves of each layer are presented with their confidence intervals in Fig 4. We also represented the narrowband equivalent $S_v$ spectra (small crosses on the spectrum at 18, 38, 70, 120, 200 and 333 kHz).

The predicted $S_v(f)$ spectrum in the surface layer (Fig 4(a)) was dominated by small siphonophores (mean pneumatophore size 0.27 mm) and *limacina* pteropods. The modeled $S_v(f)$ of

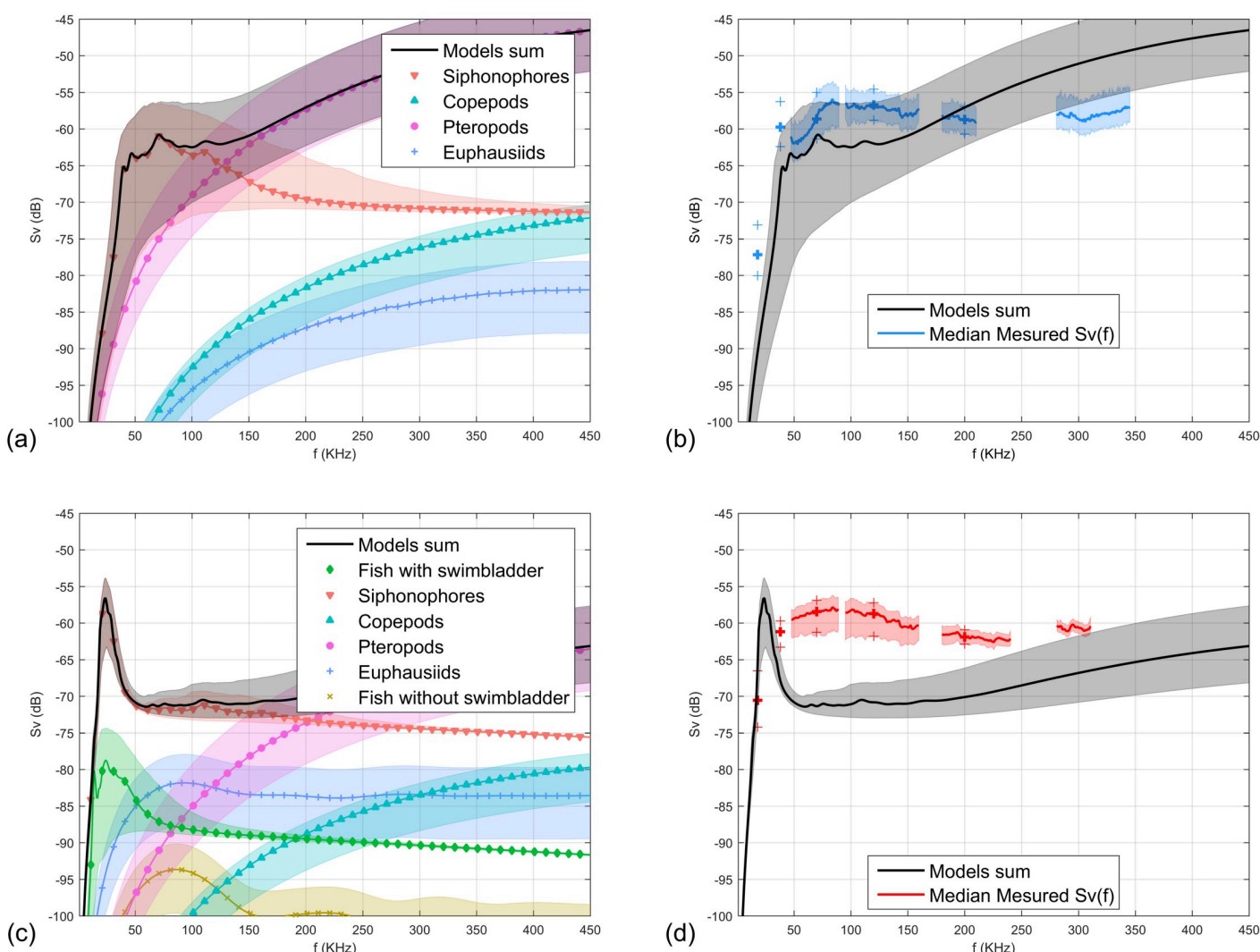

**Fig 4.** Forward approach results: modeled frequency spectra ($S_v(f)$) are represented in the left column (Fig (a) for surface layer, (c) for deep layer). Coloured areas: confidence intervals (5th and 95th percentiles of uncertainty analysis results). Measured $S_v(f)$ (red or blue lines) and predicted $S_v(f)$ (black line) frequency spectra are represented in the right column, (Fig (b) for surface layer,(d) for deep layer). Coloured areas around measured $S_v(f)$: confidence intervals (25th and 75th percentiles). Volume backscattering strength $S_v$ level at 18, 38, 70, 120, 200 and 333 kHz and their 25th and 75th percentiles confidence intervals are represented by crosses.

small siphonophores displayed a widely spread resonance peak between 38 and 150 kHz. At low frequencies (18-160 kHz) the modeled siphonophore contribution roughly followed the measured $S_v(f)$ shapes although 3-4 dB lower in level. However, the peak at 38 kHz detected in the measured $S_v(f)$ was not observed in the modeled $S_v(f)$. Futhermore, the predicted $S_v(f)$ was 5 dB higher than the measured $S_v$ for frequencies greater than 225 kHz due to the high contribution of pteropods (Fig 4(b)). Of note, narrowband frequency responses did not capture peaks at 38 and 80 kHz, which would have potentially lead to misinterpretation, had narrowband acoustics been used alone.

The predicted $S_v(f)$ of the deep layer (Fig 4(c)) was dominated by pteropod and large siphonophore (mean pneumatophore size 1.23 mm) backscatter. The siphonophore and fish resonant peaks occurred between the 18 and 38 kHz narrowband point. Importantly, predicted and measured $S_v(f)$ were close at frequencies below 38 kHz. At higher frequencies, we observed large discrepancies between both curves, with non-overlapping uncertainty. Indeed, the frequency content between 38 and 250 kHz on the measured frequency spectrum, similar to those produced by the small siphonophores in the $S_v(f)$ surface layer (Fig 4(a)), could not be reproduced based on our biological sampling data. Notice that in this case, narrowband and broadband frequency spectra displayed similar shapes.

## Discussion

In this paper, the composition of two SSLs detected in the Bay of Biscay in springtime was characterized by jointly analyzing acoustic and groundtruthing data. In doing so, fine scale heterogeneity within SSLs was evidenced and its importance for sampling was highlighted. The SSLs were generated by a variety of scatterers belonging to several taxa. Their frequency responses from 18 to 100 kHz were dominated by GB siphonophores. As reported in previous studies [1, 2], the resonance frequency of small GB organisms fall within the range of frequencies commonly used in fisheries acoustics, namely at lower frequencies (18-50 kHz). Our study confirms that small GB organisms can dominate the total backscattering at these frequencies, even at low density. Pteropods were the main contributors at higher frequencies ($> 100$ kHz), with very high densities recorded in the surface SSL (4000-5000 $ind.m^{-3}$). To our knowledge, this is the first mention of such high pteropod density.

The unsupervised clustering of broadband echograms revealed a high frequential heterogeneity within the SSLs. SSLs were stratified vertically, with mild horizontal heterogeneity at our sampling scale. Interestingly, the surface layer was composed of two sub-layers indicating a potential shift in the siphonophore size composition. These two sub-layers were sampled differently by the nets. Two sub-layers out of three were sampled in the deep layer: one had a flat $S_v(f)$ spectrum that could be attributed to gaseous scatterers, and the other showed a bump in its $S_v(f)$ curve which not be assigned to any known scatterer. Unfortunately, these SSL frequency heterogeneities could not easily be assessed during the survey by simply looking at the pulse compressed echograms displayed in real time. Hence, net sampling targeted at best the larger scale features of SSLs. The lack of real-time information on these SSL fine scale frequency heterogeneities thus hindered the characterization of their fine scale composition.

Despite some uncertainty in the net positioning in the surface layer, the measured and predicted $S_v(f)$ matched reasonably well in the forward approach, when considering the uncertainties around the two median curves. The two dominant scatterers producing these SSLs were likely siphonophores and pteropods. However the size diversity of siphonophores is suspected to be greater in the water than in the nets, as a peak at 38 kHz in the measured $S_v(f)$ remained unexplained by the catches. Additionally, some of the mismatch observed at higher frequency ($> 200$ kHz) might be explained by complex scattering [52] due to the high density

of pteropods, or by a non-random distribution of pteropods in their position or angle (e.g. because of aggregation due to mucus or opercular opening in the beam direction).

The physical vertical gradients were weak in the study area, and the surface backscattering at 18 kHz was low (-77 dB) relatively to higher frequencies. We therefore neglected the potential backscattering caused by physical phenomena such as microstructures [2] in the surface layer, as microstructures normally present a high backscattering at 18 kHz [53], and a relatively stable frequency response at higher frequencies [2].

In the deep layer, discrepancies between predicted and measured $S_v(f)$ curves could not be explained by the uncertainty in model parameters. Those differences likely resulted from the under-sampling of important scatterers. The resonance of larger siphonophores and fish juveniles was positioned between 18 and 38 kHz, while the spectrum bump at higher frequency could not be explained by the biological sampling. This could be caused by the presence of un-sampled small siphonophores, or by a hypothetical accumulation of centimetric flocks of suspended matter [54] in relation with the high turbidity (S3 Fig in the Supporting Informations).

Moreover, *Argentina sphyranea* juveniles were modeled as FL organisms. The timing of *Argentina sphyranea* juveniles swimbladder development is however unclear [51, 55]. Therefore, the modeled $S_v(f)$ of *Argentina sphyranea* juveniles might have been largely underestimated at lower frequencies if a functional swimbladder was in fact present.

The forward approach required accurate acoustic backscattering models of dominant scatterers and a good parameterization of these models. The uncertainty analysis enabled to represent the potential errors on model parameters and better assess the discrepancies between predicted and measured $S_v(f)$ curves. However, [42]'s model, which was used here, is designed for ka < 0.1, and is less reliable for a higher ka. This decrease in GB model precision at higher frequencies has low impact on the overall results, as GB backscattering was generally low at higher frequencies relatively to other scatterers. Nonetheless, it could be necessary to supplement [42]'s model with a gaseous cylinder model (e.g. [56]) when GB organism backscattering dominates the entire frequency spectrum.

The largest source of bias and uncertainty however likely resided in the biological sampling, due to avoidance [13] and fragile organism destruction by the nets issues [57]. Biological sampling errors were however not included in our uncertainty analysis, due to difficulties to quantify them. One way forward might be to use the discrepancies between the measured and predicted $S_v(f)$ curves to assess the catchability bias of the nets, when measured $S_v(f)$'s are higher than predicted ones [13].

Our results illustrate the potential of broadband acoustics to (i) fill-in the gaps between narrowbands, (ii) investigate the SSLs fine frequential heterogeneity and (iii) ascertain the presence or absence of resonance peaks [21] and derive their actual maximum backscattering intensity (Fig 4(b)). Narrow-band acoustics can lead to serious miss-interpretation when resonant peaks are present far away from the sampled frequency (Fig 4(d)).

Siphonophores, especially smaller ones, were difficult to sample, as they are partially destroyed by classical plankton nets [58]. This difficulty (here, one net out of two sampled them) can lead to an underestimation of pneumatophore size diversity and density. The use of ZooCAM imaging [35] was essential for detecting their presence and counting the small siphonophore pneumatophores, too small to be seen on *in situ* video recordings. The *in situ* video recordings were essential to ascertain the presence and count large siphonophores [59].

Importantly, the presence of physonect siphonophores was confirmed in the Bay of Biscay [24, 60], as well as their potential strong backscattering intensity from 18 to 120 kHz. This finding could explain the frequent occurrence of widely spread SSLs displaying resonance-like peaks in the Bay of Biscay during springtime ([23, 24, 27], whose origin remained unexplained.

In conclusion, the combination of broadband acoustics, nets, imagery and video presented in this paper paves the way for the characterization of composite SSLs produced by complex assemblages of scatterers, of various taxa and size class. This opens up a perspective for acoustic monitoring of the spatio-temporal dynamics of well identified SSL, which might be an essential step in a general comprehension of pelagic ecosystems in the context of global climate change.

## Supporting information

**S1 Appendix. Echosounder calibrations.**
(PDF)

**S2 Appendix. Gaseous target backscattering model.**
(PDF)

**S1 Fig. Noise spectrum.** Noise measurements (volume backscattering in decibel, color key) performed with the echosounders in passive mode during the PELGAS2017 survey. x-axis: acoustic frequency (kHz), y-axis: range from transducer (m).
(TIF)

**S2 Fig. Images of main scatterers.** Images of main scatterers in the samples taken with (a), (b) and (c): zooCAM; (d) and (e): Zooscan; (f), (g) and (h): binocular microscope.
(TIF)

**S3 Fig. Hydrological context.** Temperature (a), salinity (b), fluorescence (c) and turbidity (d) profiles. Each dashed line represent one profile, the solid lines represent the mean profile for daytime (red) and night time (blue).
(TIF)

**S1 Table. Description of used mesozooplankton and micronekton nets.** Depth and time of the net tows. Each gear was associated with a station number. The multinet open each of its nets at a specified depth, presented in net depth interval. Only the multinet nets sampling in a sampled layer are presented in this table.
(PDF)

## Acknowledgments

The authors would like to thank the R/V Thalassa Captain Loic Provost, his crew and all the scientists involved in the 2016 PELGAS surveys. We would also like to thank SIMRAD and Lars N. Andersen, for lending the EK80 echosounders. Special acknowledgements to Fabien Lombard for his priceless help on siphonophore identification, to Cyrill Gallut and Nalani Schnell for micronekton identification and Julien Simon for the use of cameras on nets. We want additionally to thank Sven Gastauer, Dezhang Chu and all the anonymous reviewers for all their pertinent comments and remarks over the manuscript.

## Author Contributions

**Conceptualization:** Arthur Blanluet, Mathieu Doray.

**Data curation:** Laurent Berger, Jean-Baptiste Romagnan, Naig Le Bouffant.

**Formal analysis:** Arthur Blanluet.

**Funding acquisition:** Pierre Petitgas.

**Investigation:** Arthur Blanluet.

**Methodology:** Arthur Blanluet, Sigrid Lehuta.

**Project administration:** Arthur Blanluet, Mathieu Doray, Laurent Berger, Pierre Petitgas.

**Resources:** Mathieu Doray, Laurent Berger, Pierre Petitgas.

**Software:** Laurent Berger, Naig Le Bouffant.

**Supervision:** Mathieu Doray, Laurent Berger, Pierre Petitgas.

**Validation:** Mathieu Doray, Laurent Berger, Jean-Baptiste Romagnan, Naig Le Bouffant, Sigrid Lehuta, Pierre Petitgas.

**Visualization:** Laurent Berger, Jean-Baptiste Romagnan, Pierre Petitgas.

**Writing – original draft:** Arthur Blanluet.

**Writing – review & editing:** Mathieu Doray, Laurent Berger, Jean-Baptiste Romagnan, Pierre Petitgas.

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
