## [Decision Letter · Decision Letter 0]

9 Aug 2019

PONE-D-19-17373

Characterization of sound scattering layers in the Bay of Biscay using broadband acoustics, nets and video

PLOS ONE

Dear Dr. Blanluet,

Thank you for submitting your manuscript to PLOS ONE. After careful consideration, we feel that it has merit but does not fully meet PLOS ONE’s publication criteria as it currently stands. Therefore, we invite you to submit a revised version of the manuscript that addresses the points raised during the review process.

We would appreciate receiving your revised manuscript by Sep 23 2019 11:59PM. To enhance the reproducibility of your results, we recommend that if applicable you deposit your laboratory protocols in protocols.io, where a protocol can be assigned its own identifier (DOI) such that it can be cited independently in the future. For instructions see: http://journals.plos.org/plosone/s/submission-guidelines#loc-laboratory-protocols

We look forward to receiving your revised manuscript.

Kind regards,

Stavros Ntalampiras

Academic Editor

PLOS ONE

Journal Requirements:

A. Blanluet doctoral grant was co-funded by the Conseil R ´egional des Pays de la Loire and IFREMER.

Reviewers' comments:

Reviewer's Responses to Questions

**Comments to the Author**

1. Is the manuscript technically sound, and do the data support the conclusions?

Reviewer #1: Yes

Reviewer #2: Yes

2. Has the statistical analysis been performed appropriately and rigorously? 

Reviewer #1: N/A

Reviewer #2: Yes

3. Have the authors made all data underlying the findings in their manuscript fully available?

Reviewer #1: No

Reviewer #2: Yes

4. Is the manuscript presented in an intelligible fashion and written in standard English?

Reviewer #1: Yes

Reviewer #2: Yes

5. Review Comments to the Author

Reviewer #1: First of all, for question 3 above I found some 2013 PELGAS data (475 MB) at

DORAY Mathieu, DUHAMEL Erwan, HURET Martin, PETITGAS Pierre, MASSE Jacques (2000) PELGAS, https://doi.org/10.18142/18,

but I never located the 2016 data analyzed in this paper.

I feel that the paper is publishable. It compares broadband acoustic backscatter data collected from a ship with expectations based on net sampling and forward modeling of scattering strength. The comparison shows some discrepancies between the two that illustrate the challenges of both types of sampling. The acoustic data show some informative patterns.

First of all, the paper cites important and needed scientific work in the area. The net collection and forward modeling is explained. I have only a few other comments on the paper, mostly minor.

There is a font problem with greater than and/or less than symbols, as at line 11, line 163, line 341, line 367, maybe other places.

The sentence at line 15 could be changed to begin 'Uncertainty of the species composition in the SSL's ...

The resonance discussion (line 39) should mention that there is a frequency of energetic response related to the vibrational characteristics of the scatterer, the resonant frequency. The term 'given frequency' doesn't really convey any useful information. Also resonance is not restricted to gaseous inclusions.

Line 104: Do you mean fluorometer?

Page 7: I could not find what EI means.

page 11: I could not determine the reason for making Sv standardized. Is this a function of frequency? If Sv(f) is from calibrated systems, what is the reason for adding on the quantity that is an average over the transducers? (N=6 transducers, correct?)

The language of the sentence at line 364 needs repair. I don't understand the statement. This is true at line 414 also, in S1 Appendix.

Reviewer #2: This is one of a few high quality manuscripts I have reviewed over decades of my professional career. The field experiment was carefully designed and the co-registered acoustic, biological, optical, and oceanographic data were collected properly. Analysis of collected data including the scattering modeling and uncertainty assessment is scientifically sound and very objective. I recommend it be published with some minor revisions suggested in the attached file.

6. PLOS authors have the option to publish the peer review history of their article (what does this mean?). If published, this will include your full peer review and any attached files.

Reviewer #1: No

Reviewer #2: Yes: Dezhang Chu

---

## [Author Response · Author response to Decision Letter 0]

13 Sep 2019

Reviewer 1: I have incorporated all your suggestion to my revision. Thank you for your help. 

Reviewer 2: I have incorporated all your suggestion to my revision. Thank you a lot for your helpful comments and your encouragement!

---

## [Decision Letter · Decision Letter 1]

25 Sep 2019

Characterization of sound scattering layers in the Bay of Biscay using broadband acoustics, nets and video

PONE-D-19-17373R1

Dear Dr. Blanluet,

We are pleased to inform you that your manuscript has been judged scientifically suitable for publication and will be formally accepted for publication once it complies with all outstanding technical requirements.

With kind regards,

Stavros Ntalampiras

Academic Editor

PLOS ONE

Additional Editor Comments (optional):

Reviewers' comments:

Reviewer's Responses to Questions

**Comments to the Author**

1. If the authors have adequately addressed your comments raised in a previous round of review and you feel that this manuscript is now acceptable for publication, you may indicate that here to bypass the “Comments to the Author” section, enter your conflict of interest statement in the “Confidential to Editor” section, and submit your "Accept" recommendation.

Reviewer #1: (No Response)

Reviewer #2: All comments have been addressed

2. Is the manuscript technically sound, and do the data support the conclusions?

Reviewer #1: Yes

Reviewer #2: Yes

3. Has the statistical analysis been performed appropriately and rigorously? 

Reviewer #1: Yes

Reviewer #2: Yes

4. Have the authors made all data underlying the findings in their manuscript fully available?

Reviewer #1: Yes

Reviewer #2: Yes

5. Is the manuscript presented in an intelligible fashion and written in standard English?

Reviewer #1: Yes

Reviewer #2: Yes

6. Review Comments to the Author

Reviewer #1: The small improvements needed in the paper have been done. There are a few places where the language is troublesome and the meaning is not absolutely clear (e.g lines 16-18) but this does not stand in the way of the science.

Reviewer #2: The figures are all in low-resolution format and may be they are only for the review purpose. If the final pdf will have full resolution, I'll satisfied with this revised manuscript and recommend its acceptance.

7. PLOS authors have the option to publish the peer review history of their article (what does this mean?). If published, this will include your full peer review and any attached files.

Reviewer #1: No

Reviewer #2: Yes: Dezhang Chu

---

## [Editor Report · Acceptance letter]

10 Oct 2019

PONE-D-19-17373R1 

Characterization of sound scattering layers in the Bay of Biscay using broadband acoustics, nets and video 

Dear Dr. Blanluet:

I am pleased to inform you that your manuscript has been deemed suitable for publication in PLOS ONE. Congratulations! Your manuscript is now with our production department. 

With kind regards,

on behalf of

Prof. Stavros Ntalampiras 

Academic Editor

PLOS ONE